# Unusual hydrogen implanted gold with lattice contraction at increased hydrogen content

Khac Thuan Nguyen [1], Van Hiep Vuong [2], The Nghia Nguyen[2], Trong Tinh Nguyen[3], Tomoyuki Yamamoto [4] & Nam Nhat Hoang [1,5✉]

The experimental evidence for the contraction of volume of gold implanted with hydrogen at low doses is presented. The contraction of lattice upon the addition of other elements is very rare and extraordinary in the solid-state, not only for gold but also for many other solids. To explain the underlying physics, the pure kinetic theory of absorption is not adequate and the detailed interaction of hydrogen in the lattice needs to be clarified. Our analysis points to the importance of the formation of hydride bonds in a dynamic manner and explains why these bonds become weak at higher doses, leading to the inverse process of volume expansion frequently seen in metallic hydrogen containers.

[1] Faculty of Engineering and Nanotechnology, VNU-University of Engineering and Technology, 144 Xuan Thuy, Cau Giay, Ha Noi, Vietnam. [2] Faculty of Physics, VNU-Hanoi University of Science, 334 Nguyen Trai, Thanh Xuan, Ha Noi, Vietnam. [3] Institute of Applied Physics and Scientific Instrument, Vietnam Academy of Science and Technology, 18 Hoang Quoc Viet, Ha Noi, Vietnam. [4] Kagami Memorial Research Institute for Materials Science and Technology, Waseda University, Shinjuku, Tokyo 169-0051, Japan. [5] Advanced Institute of Engineering and Technology, VNU-University of Engineering and Technology, 144 Xuan Thuy, Cau Giay, Ha Noi, Vietnam. ✉email: namnhat@gmail.com

Hydrides and their applications on a large scale have been known for a long time, e.g., Ni-metal hydride batteries. The recent technology renaissance and the surge in metal hydride studies come from their hydrogen storage capacity, which is higher than that of other types of storage[1]. Interestingly, high $T_C$ superconductivity has recently been reported for other kinds of hydrides: carbonaceous sulfur hydride (287.7 K at 267 GPa)[2], hydrogen sulfide (203 K at 150 GPa)[3,4], and lanthanum superhydride (260 K at 180 GPa)[5]. Under ambient pressure, solid metal hydrides, such as hydrogen-implanted Pd-based alloys (with Cu, Ag, and Au), also show superconductivity ($T_C \approx$ 15–17 K)[6–10]. These Pd-Au alloys are the only forms of solid hydrides containing gold that are available today at ambient pressure. The known solid gold hydrides are unstable and could be prepared only at high temperature and pressure[11]. Notably, there are many molecular forms of gold hydrides, such as AuH, AuH$_2$, (H$_2$)AuH, or in general Au$_n$H$_m$[12–16]. The structural properties of these compounds, including the bonding geometries and spectral responses, are well documented (Table 1). The ground and excitation states in AuH were studied quite early and have been presented in detail[17,18]. For the binding of gold, the relativistic effect (caused by fast moving core electrons) and the associated reduction of the 5d-6s gap play a crucial role[19–21]. Some monohydrides may also be synthesized in bulk form under high pressure, e.g., AuH (at 5 GPa and 400 °C)[11] (CuH, MgH, etc.)[22,23]. Because hydrogen exhibits a broad range of bonding abilities, it is necessary to determine the correct positions of hydrogen atoms in the host lattice to reveal the bonding geometries. Unfortunately, this information is not often available. Due to the low scattering

power of hydrogen, direct measurements using X-ray and neutron scattering techniques are limited[24]. On the other hand, the data from nuclear magnetic resonance (NMR), rotational spectroscopy, are available only for the metal-organic compounds of Au and other metals (e.g., Cu, Pt, and Pd)[25–28]. To date, for solid gold hydrides, we have been limited to the data obtained from spectroscopic methods (IR, UV-Vis, Raman). These data can be compared with those of molecular hydrides, which are again retrieved from the spectral responses and interpreted with the help of computation methods[12–16,29–33] (Table 1). From the few data available, one can depict the metal-to-hydrogen bonding in molecular gold hydrides as the covalent type, where the Au atoms (with an electronegativity of 2.54 on the Pauling scale) are negatively charged and hydrogen atoms (with an electronegativity of 2.2 on the same scale) are positively charged. The projected Mulliken charges above (Au, H) pairs vary from case to case and are roughly equal 0.2e, where the Au-H bond lengths average at 1.63 Å. The question therefore arises as to whether this figure also applies to metallic gold hydrides Au$_n$H$_m$ at dilute hydrogen concentrations ($n \gg m$), or whether there is another type of dynamic hydride bonding in the lattice due to the extreme light mass and high hydrogen mobility. There are many studies on dissolved hydrogen in fcc metals, e.g., in Pd, Cu, Al, but those on hydrogen in gold or gold alloys are very limited[34–41]. None of the studies report lattice contraction. It is known that α and β hydrogen phases exist in metals. The first relates to an exothermic solid solution of atomic hydrogen at low concentrations and the second corresponds to the formation of hydrides at high hydrogen contents[42]. The word "hydride" refers to defined molecular forms within metals. Phase α may also exist in liquid metals at moderate hydrogen pressure, e.g., in Pd about 1%[43]. Attention is drawn to the report on the reduction of resistivity by H$_2$ adsorption on Au thin film by 0.5 μm[34]. The authors tried to explain this effect by reducing the lattice strain induced by H$_2$ chemisorption. However, the effect of the lattice strain itself on the resistance is not clear and, in addition, hydrogen is not present in the metals in molecular form[44], so the decrease in resistivity by introducing hydrogen requires a deeper explanation. Another important observation is the absence of lattice expansion in H charged Al[45,46]. This effect is attributed to the existence of a large number of vacancies created during hydrogen recharge. Using the method and data provided in these articles and in reference[47] we estimated the concentrations of vacancies in our samples of the order of 5 to 27%. These values are compatible with that of Pd-H (18%) but are large compared to a value estimated by TRIM program[48] for a 100 keV proton beam at 1 μm Au target (0.9%). In this study, we present experimental data that show the contraction of the lattice of gold thin films under hydrogen implantation and provide a theoretical analysis of the dynamic bonding of hydrogen to the metallic lattice. With this type of bond, gold thin films implanted with H, stable under ambient conditions, can be considered a new type of compound.

### Results

Until now, the only cases of metal hydrides prepared by means of an ion implantation technique are the superconducting Pd (Cu, Ag, Au) alloys mentioned above. These alloys, exhibiting large hydrogen contents (from ~30 to 100%, i.e., from 6 to 19 × 10$^{17}$ hydrogen atoms per cm$^2$) were prepared and measured at $T < 20$ K to prevent the melting of samples due to the high current density of the applied ion beams[6–9]. Therefore, it is not clear whether these alloys are stable under ambient conditions. Conversely, to obtain stable films under ambient conditions, we processed the implantation at a much lower current density (on the scale of nanoampere per cm$^2$) and kept all experiments at room temperature. The details of preparation are given in the

**Table 1 Bonding in gold hydrides.**

| Formation | Bond length (Å) | Frequencies (cm$^{-1}$) | Ref. |
|---|---|---|---|
| Experimental and calculated data for molecules | | | |
| AuH | 1.546[a] | 2227.4[a], 2226.6 | 14,30 |
| AuH$_2$ | 1.619[a], 128.6[a] | 1995.2[a] | 14 |
| Au$_x$H | — | 2076, 1986 | 14 |
| (H$_2$)AuH | 1.574[a] | 2005.9[a], 2173.6 | 14 |
| Au-H/Au/CeO$_2$ | — | 2125 | 33 |
| Au-H/Au/ZrO$_2$ | — | 1621 | 29 |
| Au-H/Au/CeO$_2$ | — | 1800, 2126 | 29 |
| (H$_2$)AuH | 1.577 | 2164.0 | 30 |
| (H$_2$)AuH$_3$ | 1.646 | 1661.5 | 30 |
| (AuH$_2$)$^-$ | 1.668[a] | 1636.0 | 30 |
| (AuH$_4$)$^-$ | 1.653[a] | 1676.4 | 30 |
| AuH | 1.541 (1.546)[b] | 2271 (2223)[b] | c |
| (AuH$_2$)$^-$ | 1.649 (1.657)[b] | 1753 (1692)[b] | c |
| (AuH$_4$)$^-$ | 1.639 (1.644)[b] | 1815 (1759)[b] | c |
| Au$_2$H | 1.682 (1.687)[b] | 1491 (1403)[b] | c |
| Calculated data for periodic structures and clusters | | | |
| $I mm2$(Au$_4$H)[d] | 4 × 1.954 | 352 | c |
| $F\bar{4}3m$ (Au$_4$H) | 4 × 1.864 | 992 | c |
| $R3m$ (Au$_3$H) | 3 × 1.835 | 241, 837 | c |
| $Pmm2$ (Au$_2$H) | 2 × 1.782 | 444, 799, 935 | c |
| $I mmm$ (Au$_2$H) | 2 × 1.733 | 1381 | c |
| $P1$ (Au$_2$H) | 1.849, 1.869 | 315, 778, 1113 | c |
| 3D cluster (Au$_6$H) | 2 × 1.821 | 390, 736, 1243 | c |
| 2D cluster (Au$_{16}$H) | 2 × 1.754 | 375, 881, 1385 | c |
| Observed data for implanted gold thin films | | | |
| $P1$ (Au$_n$H) | --- | 460, 815, 910, 1080, 1125, 2100 | c |

[a] Calculated data for molecules at the B3LYP level.
[b] LDA (GGA) calculated data using large supercells.
[c] This work.
[d] Corresponding bonding configuration is given in the parentheses.

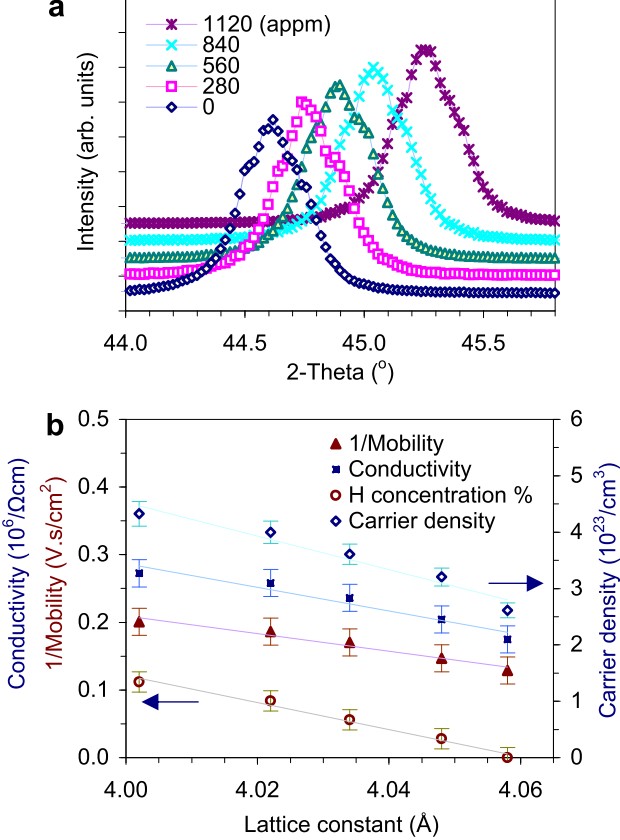

**Fig. 1 Hydrogen-implanted gold in the diluted region. a** The shift towards higher 2-theta values of the highest diffraction peaks due to increased hydrogen content (0, 280, 560, 840 and 1120 appm) is observed. The SEM images of surfaces of a sample with 560 appm (100 µC) hydrogen content before and after implantation are shown in Supplementary Fig. 1. **b** The dependences of the conductivity, carrier density, carrier mobility and H concentration (%, i.e., $10^4$ appm) on lattice constant. The lines are drawn to only guide the eyes. Error bars indicate standard deviations.

Methods section. To our great surprise, there is a systematically large contraction of the lattice constants due to implantation, while the symmetry remains *fcc*, which has not changed since the symmetry of gold (Fig. 1a). This observation contrasts with all previously reported results, as lattice expansions have always been detected[22,23]. For comparison, while our sample with a 0.11% implanted hydrogen content (1120 appm) induces a 4.4% contraction of the unit cell volume, a sample reported previously with a 75% H-doped content shows a volume expansion of 12%[22,23]. The contraction of the lattice was not the only unusual deviation from common sense that was observed. Indeed, an obvious increase in the room temperature conductivity due to increased hydrogen content was detected (Fig. 1b). This finding follows hand in hand with the increase in carrier density and decreased carrier mobility, and is difficult to explain from the gradual generation of vacancies caused by ion bombardment. All samples appear to be metals at room temperature, with conductivities of approximately $10^6$ times greater than that of the samples with a 90% H-doped content, as previously reported[6]. It is quite intriguing that the implanted samples possess higher carrier densities and better electrical conductivities. Indeed, for the heavily doped samples previously reported[22,23], a drop in the conductivity due to implantation was observed, and the samples were not metallic at room temperature.

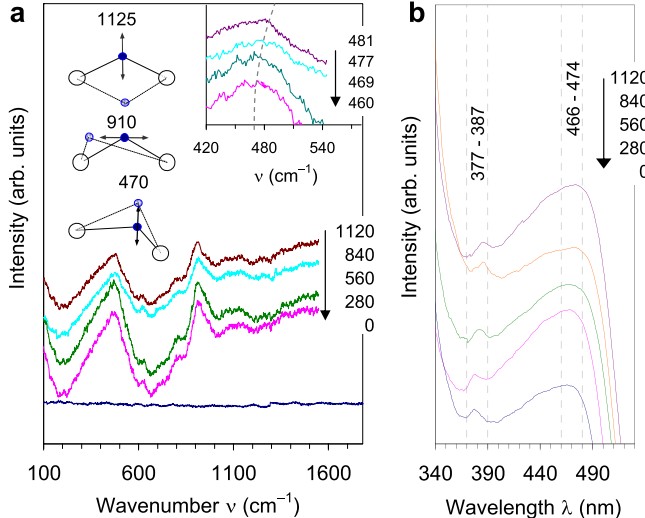

**Fig. 2 Optical spectra of the implanted samples. a** Raman scattering spectra with assignments for the 3 $A_{1g}$ modes; the inset shows the red shifts of the peaks at 460 cm$^{-1}$. **b** UV-Vis absorption spectra.

**Raman measurement.** The presence of hydrogen is clearly indicated in the Raman scattering spectra (Fig. 2a). While the scattering features are absent for pure gold, the hydrogen-implanted films show two strong bands at 460 and 910 cm$^{-1}$, together with weaker ones at 815, 1091, 1125 and 2100 cm$^{-1}$. High-frequency features were also reported previously for molecular hydrides, e.g., 2164[14,30] and 2125 cm$^{-1}$ [33] (Table 1). These frequencies correspond to a strong one-to-one binding of hydrogen in the AuH molecule (Au-H distances <1.58 Å). The frequency decreases with increasing binding distance, where the hydrogen atom binds simultaneously to 2, 3 or 4 gold atoms. To understand the Raman spectra in a periodic lattice, where the site symmetry is important, recall that in the space group $Fm\bar{3}m$, the origin (0,0,0) and the face-center positions do not imply any Raman or IR active modes, except for the low energy $T_{1u}$ acoustic phonons (this is why IR and Raman spectra are absent for the pure bulk gold, Fig. 2a). However, the general positions $(x,y,z)$ may imply different IR and Raman active modes. The final mechanical representation is given as follows[49]:

$$M = 9T_{1u} + 3A_{1g} + 6E_g + 9T_{2g} \tag{1}$$

Except for the acoustic modes and the weak $E_g$ and $T_{2g}$ modes, the 3$A_{1g}$ modes can be clearly observed. All Raman active modes are also IR active, which is why the IR and Raman spectra are not very different from each other even for the molecular hydrides. Using GGA/PBE functionals, the Raman spectra are simulated by means of density functional theory (DFT) for various space group and site symmetry settings. The relaxations of the lattices with H at different positions lead to different final symmetries as follows: (i) Au$_4$H in $F\bar{4}3m$ or $Imm2$ (4 × 1.954 Å): one H binds to 4 gold atoms of equal bond lengths, 1 mode at 352 cm$^{-1}$; (ii) Au$_3$H in $R3m$ (3 × 1.835 Å): 2 modes at 241, 837 cm$^{-1}$; (iii) Au$_2$H in $Pmm2$ (2 × 1.782 Å): 3 modes at 444, 799 and 935 cm$^{-1}$; (iv) Au$_2$H in $Immm$ (2 × 1.733 Å): 1 mode at 1381 cm$^{-1}$; and (v) Au$_2$H in $P1$ (1.849, 1.869 Å): 3 modes at 315, 778 and 1113 cm$^{-1}$ (Table 1). The simulation also confirms that there is no active mode where H occurs at the *bcc* position in the $Fm\bar{3}m$ lattice. Different bonding configurations are due to different local hydrogen symmetries and naturally resonances originating from different sets of Au-H bonds contribute differently to phonons. From this analysis, the assignment of the observed modes

(Fig. 2a) is straightforward. The weak features at 1125 cm$^{-1}$ correspond to the symmetric stretching mode, where the hydrogen atom is moving along the perpendicular bisector of the equilateral triangle Au-H-Au. The clear peaks at 910 cm$^{-1}$ correspond to the anti-symmetric stretching mode, where the hydrogen atom is moving along a parallel direction to the Au-Au bond. The main feature observed at 460 cm$^{-1}$ results from the symmetric displacement mode, where the hydrogen atom is moving in a direction perpendicular to the Au basal plane. The changes in each mode position result from different site symmetries of hydrogen in different space groups.

**Phonon dispersion and negative lattice expansion.** From the measured Raman spectra, the mode Grüneisen parameters defined as $\gamma_i = -\frac{V}{\omega_i}\left(\frac{\partial \omega_i}{\partial V}\right)$ can be calculated directly. The red shifts are noticeable for the two symmetric modes at 460 and 1125 cm$^{-1}$ (inset, Fig. 2a), but they are less pronounced for the anti-symmetric mode (910 cm$^{-1}$). For structures with unchanged symmetry as the cell volume increases, the frequencies of the modes usually decrease due to the weakening of the force constants, resulting in positive Grüneisen parameters. However, for structures with decreasing symmetry during volume increase, different axis deformations can lead to shortening of some axes, which in turn cause the modes to be split into those whose frequencies increase inducing the negative mode Grüneisen parameters $\gamma_i$. This situation is very typical for hydrogen-implanted gold when local symmetry changes from $m\bar{3}m$ (hydrogen at bcc positions) to $\bar{4}3m$ (H at (1/4, 1/4, 1/4)) and mmm (0,0,1/2) and 1 (H in P1 at general positions, see Supplementary Table 1). Loss of symmetry due to hydrogen insertion usually occurs when the axis connecting the two Au atoms adjacent to the hydrogen is shortened. This loss implies the split of the associated TO$_1$ and TO$_2$ phonons that coincide with the higher symmetries. To understand the anharmonicity that emerges, we show the phonon dispersion curves obtained for selected symmetries Cmmm and P1 in Fig. 3. The magnitudes of the mode Grüneisen parameters are distinguished by colors; specifically, the brighter ones (yellow) correspond to the larger negative values, and the darker ones (red) correspond to the same but positive values. Because in Cmmm (Fig. 3a) the hydrogen atom lies along the z axis, it can either vibrate along z (asymmetric mode) or in the perpendicular directions to z (two symmetric modes). It is clear that the positive mode Grüneisen parameters are seen along the T-Y direction (that is, a direction parallel with −z), whereas the negative parameters are seen along the directions Y-G, G-S, R-Z, Z-T ($\perp z$), S-R and G-Z (// + z). Figure 3b shows that the loss of symmetry in P1 induces almost all TO-mode parameters to be negative due to the heavy splits of the TO$_1$ and TO$_2$ modes. However, the LO modes have positive values in the Q-Z and Z-G directions. Looking at the corresponding Brillouin zone vectors, these two directions are perpendicular to the Au basal planes. The analysis of eigenvectors shows that the upper phonon band (TO$_1$) corresponds to symmetric vibrations of hydrogen in its basal plane formed by Au atoms, while the lower TO$_2$ corresponds to motions perpendicular to the Au basal plane, and the LO phonon band corresponds to hydrogen vibrations in a direction parallel to the axis connecting two Au atoms near hydrogen. To further illustrate the hydrogen bonding, the electron density map is shown in Supplementary Fig. 3. The negative mode Grüneisen parameters can also be observed for the phonon modes of gold atoms. The analysis of displacement vectors reveal that the anharmonicity can arise from gold sites that are not directly linked to hydrogen. Therefore, the formation of a hydride bond

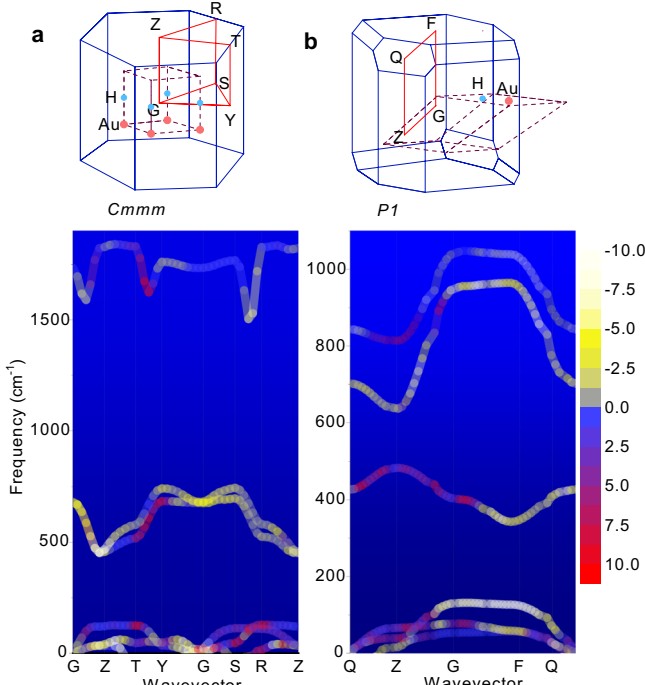

**Fig. 3 Phonon dispersion curves for selected symmetries. a** Cmmm and **b** P1. The brighter (yellowish) colors show the larger negative mode Grüneisen parameters, whereas the darker ones (reddish) correspond to the larger positive values. The Brillouin zone (blue) path (red lines) with reciprocal lattice points and the real lattice (dash brown lines) with positions of Au and hydrogen are also given. For easier comparison with the observed Raman data the phonons are given in the unit of cm$^{-1}$ instead of the conventional unit of THz.

should not be considered as the local effect limited to a given site of hydrogen, but rather in a dynamical manner, so that the non-localized nature of this bond is a primary cause of observed phonon anharmonicity. In addition to phonon anharmonicity, the optical absorption of hydrogen implanted samples expresses another interesting situation when red shifts are observed in the UV-Vis absorption spectra. The shifts are small but recognizable, i.e., 377, 378, 382, 385 and 387 nm for the first (smaller) peak and 466, 468, 469, 472 and 474 nm for the second (broader) peak. In the UV-Vis spectra, the red shifts are associated with the weakening of the oscillator strength; this needs an explanation because when the lattice is compressed, the oscillators are usually strengthened. This behavior is another interesting expression of diluted gold hydrides.

## Discussion
To clarify the physics of hydrogen bonding in the gold lattice, we performed structure optimization using the CASTEP code[50] for large supercells, i.e. for continuous structures with no vacuum slabs, where one hydrogen atom is inserted, by using the plane-wave (PW) basis with the local density approximation (LDA) functional (the detailed settings are given in Methods section and Supplementary Information). The supercells are created from the optimized $Fm\bar{3}m$ primitive cells and are denoted as 111, 222, 333… nnn, according to the number of cells in each direction (Supplementary Fig. 2). Therefore, the number of Au atoms in the nnn-supercell is $n^3$, e.g., there are 729 Au atoms in a 999 supercell. After the insertion of one hydrogen atom, the symmetry was

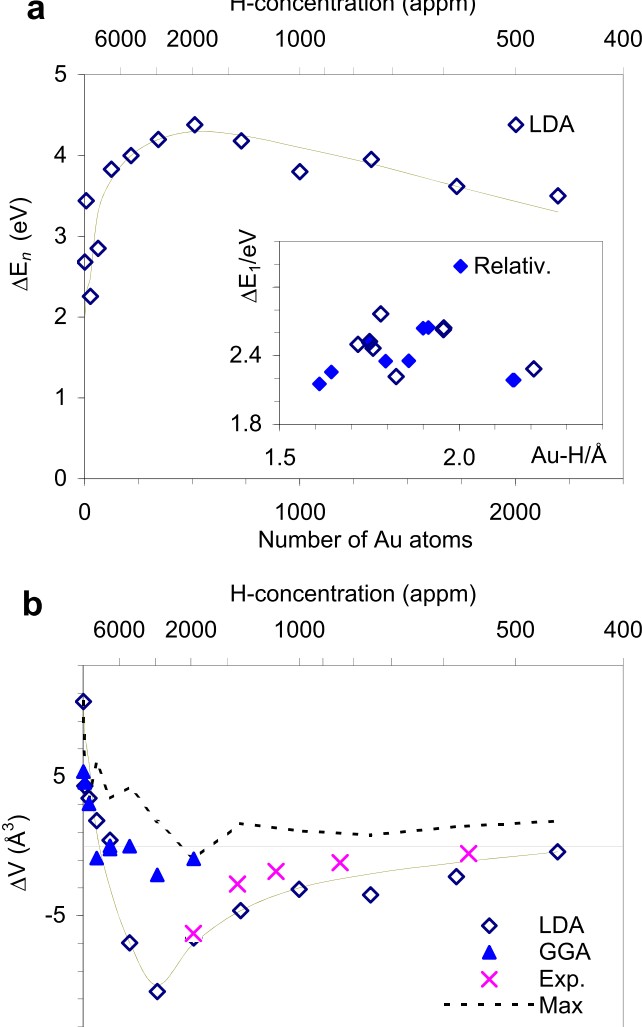

**Fig. 4 Results of the structure optimization of large supercells. a** The LDA cohesive energy as a function of the hydrogen concentration (supercell size). The inset compares $\Delta E_1$ obtained with and without a relativity correction for the case of mono gold hydride AuH. **b** The LDA derived maximum positive and negative changes of the unit cell volume as the hydrogen concentration (supercell size) changes. The data obtained by GGA/PBE are also shown, but other data are omitted for clarity.

reduced, so we optimized the supercells in the highest symmetry available, that is, in $F\bar{4}3m$ (s.g. #216) for most cases. The nominal H-concentration is straight from the size of each supercell, i.e., $100/n^3$ %, or $10^6/n^3$ appm. Obviously, dilute concentrations require large supercells for modeling. The available theoretical studies of the interaction of hydrogen with metal surfaces usually take into account hydrogen in tetrahedral ($\bar{4}3m$) or octahedral ($m\bar{3}m$) site or both[37–41,51,52]. These studies implement either slabs (i.e. periodic lattices of limited size with embedded vacuum layers) or clusters of less than 100 atoms. It is clear that these models cannot be applied for the hydrogen trapped deep in a large lattice of thousands of surrounding atoms, that is, with the case of diluted concentrations, where interactions between hydrogen atoms are negligible and the formation of hydride precipitates is not possible. The energies reported in these studies

depend to a large extent on the models used; the largest hydrogen binding energy is 2.27 eV[37]. Our results show that, for the metal hydride AuH in $Fm\bar{3}m$ with H at $bcc$ position, the optimized cell becomes large with $a = 3.12$ Å, which corresponds to a Au-H distance of 2.200 Å. The correction of the relativistic effect leads to a small reduction of Au-H = 2.153 Å (Fig. 4a), but compared to pure gold ($a = 2.83$ Å) these values are still very large. Application of 100 GPa would lead to $a = 2.79$ Å, which is close to 2.78 Å reported for bulk AuH[22,23]. The insertion of H led to a cohesive energy (defined as $\Delta E_{n,m} = E(Au_nH_m) - [E(Au_n) + mE(H)]$) of $\Delta E_1 = 2.13$ eV ($m = 1$), which is the lowest value among the cases where H occurs at (0.25, 0.25, 0.25) ($F\bar{4}3m$), (0.5, 0, 0.5) ($Immm$), (0.25, 0.75, 0.5) ($I\ mm2$), etc. The highest $\Delta E_1 = 2.73$ eV is obtained from $P1$ structure, which is relaxed from $F\bar{4}3m$. The cohesive energy increases as the number of coordinated Au-H bonds decreases from 6 to 4 and 2, along with a decrease in bond lengths. For low doped structures, i.e. for large $nnn$-supercells, the cohesive energy increases with increasing concentration up to 1950 appm ($n = 8$) but tends to decrease with further increasing concentration ($n < 8$) (Fig. 4a). For all structures below 1950 appm ($n \geq 8$), the cohesive energy remains above 3.50 eV, which confirms the better stability of these structures. When relaxed to $P1$ the optimized geometries show various bonding geometries: Au$_1$H, Au$_2$H, Au$_3$H, and even Au$_4$H. It is also clear from Table 1 that the Au-H bond distances for our models (1.73-1.95 Å) are greater than the ones of known molecular hydrides (1.52[15] −1.646 Å[30]) and correspond to the case when H is bound to at least two Au atoms simultaneously. The simulation results show that if each hydride bond shares its electron density with the delocalized Au$_n$ 6 s cloud, then the Coulomb attraction between Au atoms increases. Therefore, a stronger bond of atoms can be expected, which in turn compresses the lattice. This process can be easily verified in 1D periodic models composed of $00n$-chains (in $P1$ symmetry). For example, the lattice constant increases by 0.26 Å (from 2.56 to 2.82 Å) when H is inserted in the $001$-chain, while it increases by only 0.11 Å (from 5.14 to 5.25 Å) in the $002$-chain with the same insertion. At $n \geq 4$, the H-inserted $00n$ chains become shorter than the H-free chains. Another way to obtain a quick and good estimate is to consider that each Au-H bond pumps $0.5e$ into the conduction band (DFT provides approximately 0.5 as one Au-H bond is ≈ 1/2 of single bond order); then, we have $1e$ for the Au-H-Au bonding configuration. This means an increase in Coulomb attraction by $1/N$ (where $N$ is the total number of electrons shared with the conduction band by all Au atoms). From the integral density of states (DOS) we see that each Au atom shares with the conduction band approximately $0.5e$, so $N = 2$ for the $004$-chain and $N = 171.5$ for the $777$-supercell. It follows directly that the addition of one H atom increases the Coulomb attraction by 50% for the $004$-chain and only by 0.58% for the $777$-supercell. If the size of the optimized H-free $004$-chain is 10.36 Å, then the hydrogen insertion reduces it to 8.46 Å, assuming a linear oscillator model ($\omega = v/r = \sqrt{k/m} \rightarrow r = v/\sqrt{k/m}$, where $k$ is the force constant, $m$ is the reduced mass, $v$ is the orbital speed and $r$ is the orbit diameter). The value obtained by DFT is 9.89 Å. Now, with LDA, the $777$-supercell relaxes to a size of 28.63 Å, and the insertion of hydrogen leads to a decrease of $\sqrt{1.0058}$ to 28.55 Å. The LDA optimized value gives 28.59 Å. Evidently, the very small changes we see here lead to technical difficulties in verifying this mechanism in large 3D structures, because the larger the number of atoms, the greater the number of degrees of freedom existing in each optimization step and consequently calculation errors accumulate quickly. Therefore, we encounter increasing fluctuation of optimized results even when repeating the same optimization with precise settings. Thus, for more than 1000 atoms, we are limited not only to the

accuracy provided by LDA functional, but also to the various results obtained in independent runs. Figure 4b shows changes in supercell volumes (ΔV) after geometry optimization, obtained for $F\bar{4}3m$ symmetry up to $n = 13$ (i.e., hydrogen concentration from $10^6$ down to 460 appm). The GGA/PBE results for concentrations greater than 1950 appm ($n \leq 8$) also exhibit fluctuations and are shown for comparison. The maximum fluctuation of ΔV is also shown. It is interesting to see that the maximum lattice shrinkage occurs at 2920 appm ($n = 7$) for different optimization attempts. This supercell contains 343 Au atoms, which is a close number of atoms experimentally reported previously for the smallest Au clusters with metallic behavior[53]. The small clusters tend to behave as molecules exhibiting different orbital related absorption bands instead of a single strong plasmon resonance. Next, we will consider the *777*-supercell as the smallest cluster providing plasma-like absorption for the interpretation of the UV-Vis spectra shown in Fig. 2b. The obtained results clearly show that there are two separate regions in H-doping: the one below 2920 appm ($n \geq 7$), where the lattice is compressed, and the other above 2920 appm ($n < 7$), where the lattice is constantly expanding due to the increased H content. Reasonably, as the number of implanted sites increases, so does the (Au$_n$ 6 s - H 1 s) intraband Coulomb repulsion. This repulsion force blocks further pumping of electrons into the conduction band, and at more than 2920 appm ($n \leq 7$) this force dominates, causing lattice expansion along with weakening of the hydride bonds (Fig. 4a). It is interesting to observe that the hydride bond energy in the highly doped region decreases to the values obtained from the previous studies involving with slabs or clusters models[37,40,41]. This repulsion force also widens the energy gap between the occupied Au$_n$ 5d and 6 s states and raises the highest occupied 6 s level. Notably, the hydride bond energy is equal to a gap between the H 1 s state and the highest occupied Au$_n$ 6 s level. When this gap increases, the hydride bond cannot be formed. This behavior is why the hydride bonds can only be clearly observed in the diluted region.

We now turn our attention to the UV-Vis spectra. Gold nanoparticles show surface plasmon resonances of frequencies depending on particle size. Red shifts occur when the particle size increases and are especially large for the longitudinal mode of rod-shaped particles[54]. Microscopically, this behavior is associated with a longer turn-around time for electrons to circulate in larger orbits when particle size increases. Although such an effect does not exist in bulk gold, a similar increase in plasma size may also explain the red shifts observed in the UV-Vis spectra in our case. Figure 2b clearly shows that (i) the depicted frequencies correspond to the actual size of the Au$_n$ 6 s cloud plasma and (ii) this plasma size increases due to the increase in H content. First, consider a supercell with 343 Au atoms ($n = 7$) as the smallest cluster where plasma is formed[53]. Therefore, the size of this cluster may correspond to the actual size of the plasma in the gold films. We assume that the addition of one hydrogen atom will increase the size of this plasma by an amount proportional to the extension of the Au-H-Au bond length to the Au-Au distance. The available structural data provide a rough estimate of the relative increase of 0.47%. For the particular case of a peak at 466 nm (0 μC, undoped sample), the imposing of ion charge of 50 μC implies an implanted hydrogen content of 280 appm. This concentration can be modeled by placing 1 H atom into a *15-15-15* supercell (3375 Au atoms, 300 appm), leading to an expected red shift of 466*0.47% = 2.2 nm and a final position of 468.2 nm. Similarly, 100, 150 and 200 μC implanted samples can be modeled by *12-12-12-*, *11-11-11-*, and *10-10-10-* supercells (580, 750 and 1000 appm, or 1.9, 2.5 and 3.3-times greater concentrations), yielding the expected weakening of oscillator strengths of 4.2, 5.5 and 7.3 nm. This finding means that the expected peak positions

are 470.2, 471.5 and 473.3 nm for the 100, 150 and 200 μC samples, respectively. When comparing these values with the observed data, the agreements achieved appear quite satisfactory. A more accurate explanation can be obtained by calculating the excited states, i.e., the first unoccupied Kohn-Sham states in DFT. From the viewpoint of band structure theory, the insertion of hydrogen in the gold lattice opens the gaps between the conduction band and the lowest unoccupied bands. As can be seen in Fig. 5a, across the Fermi level are the bands related to the s- and d-electrons. This image is very similar to that of bulk Au, except for the gaps between the s-occupied and s-excited bands (circles). As the d-density above the Fermi level is small and the p-bands are fully occupied, the optical transitions are dominated by transitions within the s-bands. In AuH, the smallest gaps appear at the W-point (0.8, 2.2, 3.0, and 4.8 eV) and the K-point (1.1, 3.0 and 4.1 eV) of the first Brillouin zone. This result means that there is a broad band in the absorption spectra of gold hydrides from 400 to 560 nm, which diminishes towards 1500 nm. The gaps above 4 eV are also seen in the solid gold and correspond to a strong absorption of gold and its hydrides in the ultraviolet region. Figure 5b also shows that the maxima of H 1 s and Au 6 s densities coincide at approximately 6.5 eV and correspond to the formation of hydride bonds at the top of conduction band near the G-point. It is interesting to observe the change in the gap between the Au 5d- and 6s-bands at the G-point upon the application of pressure. At 100 GPa, this gap increases by approximately 1.3 eV, which is practically equal to the increase in ground state energy. The same situation also occurs with gold hydrides at increasing H content. Figure 5c shows the effect of lifting the 6s-band towards unoccupied states: the systematic narrowing of the excitation gaps (shifts of absorption maxima to longer wave lengths) at increasing concentration from 0 to 1950 ($n = 8$), 2920 ($n = 7$) and 4630 appm ($n = 6$). This is the course of red shifts observed in the UV-Vis absorption spectra (Fig. 2b). Clearly, there are two regions shown in Fig. 5c: the red shifts for concentrations below 4630 appm ($n \geq 6$) and the blue shifts for concentrations above this value ($n < 6$). Because the strength of the hydride bond decreases significantly for $n < 7$, the hydrogen atoms tend to occupy the electrostatic equilibrium positions. On the one hand, this expands the lattice, while on the other hand, the non-bonding hydrogen atoms form a network of screening potentials that increase the unoccupied states of the host matrix, thereby enlarging the transition gaps and inducing blue shifts in absorption bands.

The key to understanding the unusual behavior of gold films is the contribution of electrons from the hydrogen donor sites to the conduction band formed mainly by Au 6s electrons. Theoretically, this mechanism can also occur in other cases of doping elements less electronegative to gold, or even in other types of host matrices, such as Pd or Pt. For the case of H charged Al[45,46] mentioned above, because aluminum is less electronegative than hydrogen, the situation reverses: the formed hydride bond draws electrons from the conduction band but the outflow would be small. The computation shows that only ~0.1$e$ would be added to hydrogen. Therefore, even if hydrogen reaches the most favorable lattice sites, a weak bond means a weak effect on the lattice. The weak bond also explains why hydrogen occurs favorably at the surface of aluminum, and when it enters the bulk under external stimuli, damages often occur. These damages can cause large lattice changes that compensate for and suppress the small expansion induced by hydrogen insertion in regular lattice free spaces.

## Methods

**Ion beam parameters and settings**. An accelerator used (NEC 5SDH-2 Pelletron Tandem accelerator) is the electrostatic accelerator equipped with a proton beam.

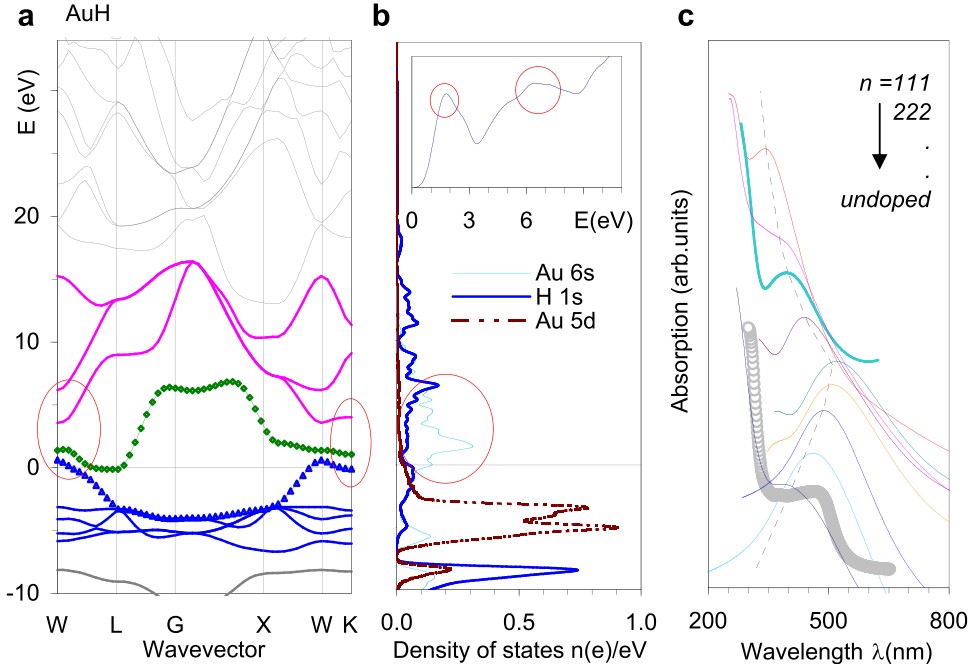

**Fig. 5 Electronic band structure and calculated optical absorption spectra of the gold hydrides. a** The band structure of the mono hydride AuH ($F\bar{4}3m$) drawn together with the graphs of the density of states (DOS) mapped onto the same energy (vertical) axis. **b** The conduction band (*d-, s-* bands) and the first 3 unoccupied bands are highlighted. The DOS-s are shown separately for Au 5*d*, Au 6 *s*, and H 1 *s* electrons to help interpretation of the band structure. The inset in the graph of the DOS shows the simulated optical absorption spectrum of AuH. **c** The calculated absorption spectra for the gold hydrides, the spectra are offset for clarity; the measured data are shown by the gray markers.

It can provide a current density of 20 nA/cm$^2$ ± 0.03%, and a maximum ion density of $9.999 \times 10^{18}$ ions/cm$^2$. The beam current density can be set to a lowest value of 1 nA/cm$^2$. The hydrogen dose is controlled by a current integrator with a typical integration accuracy of ±0.02%. The charge integration is set to 0.1 nC/pulse range from 1 to 1000 Hz by default.

**TRIM program outputs**. The TRIM program[48] shows that at 100 keV the proton beam cannot penetrate through a 700 nm thick Au film with 99.03% energy loss for ionization, 0.07% for recoils ionization, 0.25% for phonons, 1.65% backscattered, 0.03% transmitted. The vacancy generation is 8 vacancies per ion, and the ion stop range is 3928 nm with a standard deviation of 1162 nm.

**Sample implantation and characterization**. The gold thin films of thickness from 1000 to 1200 nm were prepared from Au bulk samples of purity 99.9999% by vacuum deposition technique (Univex 300). The films were implanted at room temperature with the following total charges: 0, 50, 100, 150, and 200 μC, implying the nominal concentrations of 0, 280, 560, 840 and 1120 appm. The Raman scattering measurements were carried out at room temperature on a LabRAM spectrometer equipped with a He-Ne 632.8 nm excitation laser. The UV-Vis absorption spectra were collected using a Shimadzu UV 2450 spectrophotometer. The X-ray diffractograms were recorded on a Bruker D5005 diffractometer equipped with CuK$_{\alpha 1}$ radiation (1.54056 Å); the step width was set at 0.02° for the 2-theta range from 10 to 90°. The Hall measurements were taken at 300 K and 0.56 T on a Lake Shore 8400 equipment.

**Computational settings**. LDA functional was used to optimize geometries and calculate optical properties, but where it is necessary to check the accuracy of the results obtained the GGA/PBE functional was also used. All model structures were considered as metals with spin non-polarized wave functions. The convergence tolerance for energy was set to $2 \times 10^{-5}$ eV/atom, but when higher accuracy is required, a value of $5 \times 10^{-6}$ eV/atom was used. Similarly, the maximum force tolerance was 0.05 eV/Å, but reduced to 0.01 eV/Å if necessary. By default, the minimum energy cut-off was set to 550 eV with a large k-point set range ($10 \times 10 \times 7$). For small and medium size structures, results obtained from the norm-conserving as well as from ultrasoft potentials are available. The model supercells are given in Supplementary Fig. 2. To save time and computational costs, large supercells were optimized in the highest available symmetry, which was a cubic $F43m$ with hydrogen at tetrahedral positions. But all phonon calculations were performed in $P1$ with hydrogen untied. The phonon dispersion curves were obtained using the linear response method with GGA/PBE functional.

## Data availability
All data related to the achievement of this study are given in the paper and in the Supplementary Information files.

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

## Acknowledgements

This research is funded by Vietnam National Foundation for Science and Technology Development (NAFOSTED) under grant number 103.02-2017.18. One of the authors, HNN acknowledges Prof. Le Van Hong for fruitful discussions during completion of this manuscript. This work was partly carried out at the Joint Research Center for Environmentally Conscious Technologies in Materials Science (project No. 30013) at ZAI-KEN, Waseda University, Tokyo, Japan.

## Author contributions

H.N.N. conceived the ideas, designed the experiments, analyzed data, and wrote the manuscript. N.K.T. and V.V.H. performed the preparation of materials, X-ray, Raman, UV-Vis and Hall measurements. N.T.N. performed the ion beam irradiation experiments, and N.T.T. performed the surface characterization by SEM and atomic microscope. T.Y. performed X-ray, Raman and other optical characterizations and analysis, and contributed to writing and finalizing of the manuscript.

## Competing interests

The authors declare no competing interests.
