## [Peer Review File · Nature Communications]

REVIEWER COMMENTS

Reviewer #1 (Remarks to the Author):

This study carefully demonstrated lattice contraction, instead of expansion, by small amount of hydrogen into FCC gold and provided elucidations of possible mechanisms behind the rare lattice contraction based on careful experiments with an aid of LDA-DFT calculations.

Findings in this study are of high interest from the perspective of solid state electronic structures, especially of noble metal's, chemical bonding between hydrogen and host Au atoms, possible application to minimize change in volume and, in turn, to elongate lifetime of the material, and so forth.

It is evident that lattice of FCC Au is contracted due to absorption of small amount of hydrogen, although I cannot be convinced that other factors, such as formation of Au vacancies, are excluded. The authors argues based on the analogies for Au-H molecules and attempted to interpret Raman spectra experimentally measured to explain the origin of the lattice contraction. In addition, the authors attempted to explain the lattice contraction from the viewpoint of Au-H bonding.

However, I cannot be fully convinced that the authors claim would hold, and thus I cannot recommend Editor to publish the manuscript, at least in the present form.

Major points are as follows:

1) Phonons

The authors tried to interpret Raman spectrum while doing ab initio calculations in parallel. With ab initio lattice dynamics, not only Raman-active modes but also all the modes that may be responsible for overall lattice contraction can be discussed without any difficulty. There are studies where, even if it is not lattice contraction, suppression of thermal lattice expansion by cancellation of positive and negative mode Grüneisen parameters.

2) Analogies based on molecule

The target material in this study is solid gold of which interatomic bonds and phonon modes are different from those of molecules due to electronic environment surrounding Au-H bonds. I do not understand why the authors confine the discussion within the analogies from molecules.

3) Chemical bonding between Au-H

Nowadays, ab initio calculations with plane-wave basis set offers something like Mulliken's population analysis where magnitude of bonding or antibonding of newly formed bonds can be quantified: COHP, Crystal Overlap Hamiltonian Population. I wonder why the authors keep speculating even though they have done ab initio calculations using CASTEP code.

2+3) For dynamic bonds, phonons should be discussed not only in Raman-active modes, but the authors stay within bonds in molecules and discussed with its extrapolation. For static bonds, each bond should be analyzed even in metals with unbounded electrons, but the authors discussed in rather conventional or classical way.

4) Volume change is one of primary origins of fracture or lifetime of materials when used at finite temperature and thus its study is of critical importance. If general understanding is obtained in this study with viewpoints not specific to the combination of gold and hydrogen, it can be further extended to wide spectrum of real applications. But this study remain specific to Au and H and thus it is difficult to expect that readers of this journal find interest in this form.

5) Position of H atoms

Needless to say, Pauling's electronegativity or others just state relative energies of two elements, but do not account for interatomic distance or coordination number and thus any charge transfer is possible in real circumstances. In this manuscript, it is verified that hydrogen atoms do not reside at the center of the primary interstitial site, to no surprise, taking into account the atomic radius of hydrogen. The most major question is, what is the representative position of hydrogen atoms next to gold atoms at finite temperature when atomic vibration of hydrogen atom is not harmonic.

Reviewer #2 (Remarks to the Author):

First of all I need to stress that I am a coordination chemist and not an expert in metal alloys or in structural solid state chemistry. That said, I found the observed gold lattice contraction at low H loading an interesting and unusual observation of fundamental importance that is certainly worthy of in-depth investigation. The authors have done this by a variety of spectroscopic and diffraction methods and computational modelling. The authors have presented a convincing explanation both for the contraction at low H content and the lattice expansion at higher loadings. In the opinion of this reviewer the paper is essentially suitable for publication as it stands. If there are serious objections by other reviewers with more expertise in the field, I would of course defer to their superior judgement.

Minor comments:

The readability would however be greatly improved if the manuscript was carefully edited by a native English speaker and proper use of definite and indefinite articles was introduced.

Introduction: "These Pd-Au alloys are the only forms of solid hydrides containing gold available today at ambient pressure, the gold hydrides are known to be unstable". This may be true for heterogeneous systems but a quick Google search shows that there are now a number of gold hydride coordination complexes that are perfectly stable, both thermally and chemically, see for example J. Am. Chem. Soc. 2018, 140, 8287.

p. 2: spectral methods = spectroscopic methods

p. 3 top: nano-amper = nano-ampere

Fig. 1: what are the units for hydrogen concentration?

REPLY TO REVIEWERS' COMMENTS

Dear reviewers,

Please accept our sincere thanks, below are our point-to-point responses provided with your original comments.

Sincerely,

Reviewer #1:

This study carefully demonstrated lattice contraction, instead of expansion, by small amount of hydrogen into FCC gold and provided elucidations of possible mechanisms behind the rare lattice contraction based on careful experiments with an aid of LDA-DFT calculations.

Findings in this study are of high interest from the perspective of solid state electronic structures, especially of noble metal's, chemical bonding between hydrogen and host Au atoms, possible application to minimize change in volume and, in turn, to elongate lifetime of the material, and so forth.

It is evident that lattice of FCC Au is contracted due to absorption of small amount of hydrogen, although I cannot be convinced that other factors, such as formation of Au vacancies, are excluded. The authors argues based on the analogies for Au-H molecules and attempted to interpret Raman spectra experimentally measured to explain the origin of the lattice contraction. In addition, the authors attempted to explain the lattice contraction from the viewpoint of Au-H bonding.

However, I cannot be fully convinced that the authors claim would hold, and thus I cannot recomend Editor to publish the manuscript, at least in the present form.

Thank you for the valuable comments, encouragement and positive evaluation of our research. We have read through your comments many times carefully, to ensure that we have correctly understood the problems addressed. Indeed, it was helpful for us to improve the overall quality of this manuscript by working out the answers for your stated questions.

We have examined the hydrogen implanted gold thin films during the last 4 years, with repeated experiments, preparations and measurements, several times also in the laboratories in Waseda and RIKEN, to rule out all possible random and time-dependent errors caused by varying conditions and aging of materials. What we have reported here are the results of careful experimental works during long time period.

Major points are as follows:

1) Phonons

The authors tried to interpret Raman spectrum while doing ab initio calculations in parallel. With ab initio lattice dynamics, not only Raman-active modes but also all the modes that may be responsible for overall lattice contraction can be discussion without any difficulty. There are studies where, even if it is not lattice contraction, suppression of thermal lattice expansion by cancellation of positive and negative mode Grüneisen parameters.

Thank you for this, we added a new Fig. 3, and a large part of text (2 pages), together with a new Fig.3s in *Supplementary Data*, to discuss the phonons in details. Numeric evaluations of mode Grüneisen parameters are also given in *Supplementary Data*, and the data for the graphs of phonon

dispersions are also provided in Origin 8.5 format. The phonon dispersion curves for F-43m symmetry for 222-supercell are also presented in this Origin 8.5 file, where the readers can switch on /off to see the curves (with calculated Grüneisen parameters).

In summary, we give detailed analysis of (1) the origin of negative mode Grüneisen parameters for the two cases of symmetries P1 and Cmmm; (2) the corresponding hydrogen occupation in the lattice and its vibration causing the large mode anharmonicity.

For your convenience, the Fig. 3 is reproduced here.

Fig. 3 Phonon dispersion curves for selected symmetries, (a) *Cmmm* and (b) *P1*. The brighter (yellowish) colors show the larger negative mode Grüneisen parameters whereas the darker ones (reddish) correspond to the larger positive values. The Brillouin zone vectors, the reciprocal lattice points and the real lattice with positions of Au and hydrogen are also given. For the ease of comparison with the observed Raman data the phonons are given in the unit of cm^{-1} instead of a conventional unit of THz.

As known, for the structures with unchanged symmetries during the increase of cell volume the mode frequencies usually decrease due to weakening of force constants, which results in positive mode Grüneisen parameters. However, for the structures with lowering of symmetries during the volume's increase the different deformations of axes may lead to the shortening of some axis which in turn splits modes into those whose frequencies increase, introducing the negative mode Grüneisen parameters γ_i . This situation is very typical for the hydrogen implanted gold samples when local symmetries change from $m\bar{3}m$ (hydrogen at *bcc* positions), to $\bar{4}3m$ (H at (1/4, 1/4, 1/4)) and *mmm* (0,0,1/2) and *I* (H in P1 at general positions, see Table 1s, Supplementary materials). The loss of symmetry due to hydrogen insertion happens with shortening of an axis connecting two Au atoms in the hydrogen neighborhood. This implies the split of associated TO_1 and TO_2 phonon brands which were coincided in the higher symmetries. Fig. 3 shows the phonon dispersion curves obtained for *Cmmm* and *P1*, where the magnitudes of mode Grüneisen parameters are distinguished by colors: the brighter ones (yellow) correspond to larger negative values whereas the darker ones (red) to the same but positive. By using the supercells with only one hydrogen atom inserted, the dispersion

curves split into two distinct groups of which 3 upper bands (LO, TO₁ and TO₂) belong to the motions of hydrogen, whereas the rest $3n^3$ bands belong to Au. It is useful to reveal that all Au-related phonon bands are lying below 200 cm⁻¹. In higher symmetries ($m\bar{3}m, \bar{4}3m$, not shown but available in *Supplementary data*) the mode Grüneisen parameters are all positive, but when the symmetry lowers the TO₁, TO₂ phonon bands begin to split. Because in *Cmmm* (Fig.3(a)) the hydrogen atom lies along z axis, it can either vibrate along z (asymmetric mode) or in the perpendicular directions to z (two symmetric modes). It is clear that the positive mode Grüneisen parameters are seen along T-Y direction (that is a direction parallel with $-z$), whereas the negative parameters are seen along the directions Y-G, G-S, R-Z ($\perp z$), S-R and G-Z ($\parallel +z$). Note that the directions of polarization vectors are perpendicular with that of proper eigenvectors of displacements. Fig. 3(b) also shows that the loss of symmetry in *P1* induces that almost all TO-mode parameters are negative due to heavy splits of TO₁ and TO₂ modes. But LO-modes have positive values in the Q-Z and G-Z directions. Looking at the corresponding Brillouin zone vectors, these two directions are perpendicular to the Au basal plane. The analysis of eigenvectors show that the upper phonon band (TO₁) corresponds to the symmetric vibrations of hydrogen within its basal plane formed by Au atoms, whereas the lower TO₂ with the motions in a direction perpendicular to Au basal plane, and the LO phonon band to the vibration of hydrogen in a direction parallel to the axis connecting two Au atoms in the vicinity of hydrogen.

As a conclusion, the large negative mode Grüneisen parameters appear at the directions of polarization vectors which are parallel with the axis connecting two Au atoms in the hydrogen neighborhood. The anharmonicity arises as a result of asymmetric splitting of modes due to shortening of one Au-Au axis.

For further illustration of binding of hydrogen within $F\bar{4}3m$ structure, we show in Fig. 3s, *Supplementary materials*, the map of electron density obtained for the 222-supercell, see also the discussion therein.

2) Analogies based on molecule

The target material in this study is solid gold of which interatomic bonds and phonon modes are different from those of molecules due to electronic environment surrounding Au-H bonds. I do not understand why the authors confine the discussion within the analogies from molecules.

It is totally true that bonds and phonons in solids and molecules are different, but because of its small size the hydrogen can occur in several different positions in Au host lattice, where it bonds strongly to some closest sites while remaining loose to the rest (see Fig. 3s, *Supplementary data* on how fast the force decays around hydrogen). This creates certain bonding configurations similar to that occurred in the molecules and offers analogies for comparison of bonding strengths (measured in the units of Raman shifts). Since the Raman data are not available for the solid gold hydrides but only for the molecules, we use the analogies for the ease of understanding the observed Raman spectra from the chemist's point of view. This approach may be easier for broad readers.

3) Chemical bonding between Au-H

Nowadays, ab initio calculations with plane-wave basis set offers something like Mulliken's population analysis where magnitude of bonding or antibonding of newly formed bonds can be quantified: COHP, Crystal Overlap Hamiltonian Population. I wonder why the authors keep speculating even though they have done ab initio calculations using CASTEP code.

Thank you very much, this is very interesting point. Indeed, the hydride bond in solid gold is of different nature than that of molecular hydrides, because here the hydrogen is bonding to the 6s

conduction cloud created by clusters of Au atoms. The conduction band usually contains large number of Au 6s orbitals (e.g 1000 or more) therefore the density which one hydrogen atom adds to this cloud is very small to be detected via techniques like Mulliken's population analysis. Reasonably, there rises a question of how this bond could be characterized if it could not be compared with the ones of molecular hydrides (e.g. by Mulliken charges, bond orders etc.). As the answer, we present this bond in terms of the cohesive force (Fig.3(a)) which is defined as an amount of energy released when one hydrogen atom is inserted into the gold lattice. And here the differences with the case of molecular hydrides begin, because the amount of energy released depends actually on the number of hydride bonds already presented. *There is no such correlation effect in the molecular hydrides.* As a main consequence, it follows that in the low doped region the cohesive force increases with hydrogen content but it begins to decrease at higher doses, resulting in the lattice expansion commonly seen in hydrogen containers.

2+3) For dynamic bonds, phonons should be discussed not only in Raman-active modes, but the authors stay within bonds in molecules and discussed with its extrapolation. For static bonds, each bonds should be analyzed even in metals with unbounded electrons, but the authors discussed in rather conventional or classical way.

Thank you, as mentioned above, we have rewritten the whole text to discuss phonons in details. Although this approach may be difficult for broad readers, we follow your recommendation to explain the bonding of hydrogen via phonons. As said in a conclusion, the large negative mode Grüneisen parameters appear at the directions of polarization vectors which are parallel with the axis connecting two Au atoms in the hydrogen neighborhood. The anharmonicity arises as a result of asymmetric splitting of phonon modes due to shortening of one Au-Au axis.

4) Volume change is one of primary origins of fracture or lifetime of materials when used at finite temperature and thus its study is of critical importance. If general understanding is obtained in this study with viewpoints not specific to the combination of gold and hydrogen, it can be further extended to wide spectrum of real applications. But this study remain specific to Au and H and thus it is difficult to expect that readers of this journal find interest in this form.

It is true that we do not know whether the lattice contraction upon doping will be observed in other kinds of materials or not. It is the question of future investigations. Our experimental study is restricted to a specific case of solid gold hydrides. But it can be safely confirmed that, as far as the covalent bonding holds between the donors of lower electronegativity (e.g. H, B, Si, Ge...) and the metallic acceptors of higher electronegativity, such as solid gold (or Pd, Pt...), there would possibly be valid a similar mechanism of charge transfer from the donor sites onto the conduction band which would create a same correlation effect: *bonding strength is not linear on doping concentration.* This can be verified with current modeling techniques, e.g. for hydrides of palladium (PdH) or gold borides (AuB). We can include the results for PdH in *Supplementary Data* when requested, but since there is no experimental evidence for PdH until now, we think it is not appropriate at this moment.

5) Position of H atoms

Needless to say, Pauling's electronegativity or others just state relative energies of two elements, but do not account for interatomic distance or coordination number and thus any charge transfer is possible in real circumstances. In this manuscript, it is verified that hydrogen atoms do not reside at the center of the primary interstitial site, to no surprise, taking into account the atomic radius of hydrogen. The most major question is, what is the representative position of hydrogen atoms next to gold atoms at finite temperature when atomic vibration of hydrogen atom is not harmonic.

For P1 case, the hydrogen occurs in the general positions (x, y, z) close to $(1/4n, 1/4n, 1/4n)$ (for n is the supercell size) (Table 1s, Supplementary materials). The Au atoms is at $(0,0,0)$, so H is inserted in the middle of Au-Au axis.

The obtained eigenvectors show that the TO_1 phonons correspond to the symmetric vibrations of hydrogen within its basal plane created by Au atoms, whereas the lower TO_2 phonons with the motions in a direction perpendicular to Au basal plane, and the LO phonons to the vibration of hydrogen in a direction parallel to the axis connecting two Au atoms (in the vicinity of hydrogen). So, the large negative mode Grüneisen parameters appear at the directions parallel with the axis connecting two Au atoms.

Reviewer #2 (Remarks to the Author):

First of all I need to stress that I am a coordination chemist and not an expert in metal alloys or in structural solid state chemistry. That said, I found the observed gold lattice contraction at low H loading an interesting and unusual observation of fundamental importance that is certainly worthy of in-depth investigation. The authors have done this by a variety of spectroscopic and diffraction methods and computational modelling. The authors have presented a convincing explanation both for the contraction at low H content and the lattice expansion at higher loadings.

In the opinion of this reviewer the paper is essentially suitable for publication as it stands. If there are serious objections by other reviewers with more expertise in the field, I would of course defer to their superior judgement.

Thank you very much for the acceptance of this manuscript in its present form. We tried our best to compose a manuscript that is readable for broad audience, which is one of primary criteria for this journal.

Minor comments:

The readability would however be greatly improved if the manuscript ws carefully edited by a native English speaker and proper use of definite and indefinite articles was introduced.

Thank you, the manuscript has been proof-edited by Nature.com, the certification is included with this submission.

Introduction: “These Pd-Au alloys are the only forms of solid hydrides containing gold available today at ambient pressure, the gold hydrides are known to be unstable”. This may be true for heterogeneous systems but a quick Google search shows that there are now a number of gold hydride coordination complexes that are perfectly stable, both thermally and chemically, see for example J. Am. Chem. Soc. 2018, 140, 8287.

We are sorry for this, we mean "solid gold hydrides are known to be unstable", so we have rewritten the sentence to avoid misunderstanding. We thank you again for careful reading.

p. 2: spectral methods = spectroscopic methods

p. 3 top: nano-amper = nano-ampere

Fig. 1: what are the units for hydrogen concentration?

We have corrected the mistakes. The preparation by means of Heavy-Ion technique utilizes the hydrogen concentration in micro Coulomb (μC) as the total charge of the ion beam imposed on the sample. This unit can be converted easily into the atomic ratio in percentage. We include the conversion sheet (MS Excel) in the *Supplementary Data*.

We would like to express our deep gratitude to both reviewers for their critical, valuable comments and suggestions, which helped us a lot to improve the quality of this manuscript. We also send our warm thanks to the editors for handling of the manuscript and timely communication and assistance.

Sincerely yours

REVIEWER COMMENTS

Reviewer #1 (Remarks to the Author):

The revised manuscript has addressed all the points I had previously raised and thus I suggest the editor to accept the revised manuscript for publication in Nature Communications.

Before formal acceptance, I recommend the authors to reconsider following points.

1. "relative effect" vs "relativistic effects"

As far as I understand, the latter is "more" appropriate, although the former is not incorrect. The former expression appear in P. 1, L. 32, P. 6, L. 199. Doing a google search using the double-quoted keywords with "site:nature.com" shows the latter is more related to what the authors try to mean in this manuscript.

2. Readers will find "This result explains why the bulk AuH is not stable under ambient conditions but only at high pressure" at P. 6, L. 200, be an overstatement. This result does not explain all the reason for the stabilization at increased pressure. This is a trivial issue, but I am afraid this overstatement make readers wonder if other solid statements are the same.

3. Two sequential paragraphs starting at P. 6, L. 197 may confuse many readers except for the ones who themselves frequently do ab initio calculations. The explanations are given based on the model (supercell) size, while most of the readers would understand from the opposite viewpoint: concentration of H. Thus, to avoid interupping the flow of the paper for most of the readers, I suggest the authors to reconsider the description of the supercellsize/concentration dependence of the results.

Likewise, it would be more general if the author modified the equation for the cohesive energy to,

$$\Delta E_{n,m} = E(\text{Au}_n\text{H}_m) - [E(\text{Au}_n) + mE(\text{H})]$$

Even if the authors chose to stay in the current way of explanation, explicitly specifying $m=1$ will make it easier to understand for the most of readers.

Since this appears in the beginning of Discussion and following portions are interesting, I hope the most readers give up reading the to-be-paper at this point. Although this is a trivial issue of expression, but it does affect the number of citation after publication.

4. [Major] To add one sentence each.

In order for me to have objective perspective about the statements in this manuscript, I spent a week after reading the replies to comments before reading the revised manuscript. When I read through the replies, I was immediately convinced. After a week, I could not be fully convinced when I read through the manuscript and needed back to the replies.

Thus, I suggest the authors to add one sentence for each point I raised in the previous round in the beginning or wherever to convince the reviewers that the authors have thoroughly considered the differences between hydride molecules and solid gold, the limitation and sufficiency of Raman spectral analysis in this specific case, clear distinction between static lattice expansion due to static Au-H bonds and dynamic ones due to its extremely light mass of H, and so on. And/or put a stronger pointer to Supplementary Information.

5. For better readability

Paragraph newly added in the revised manuscript starting at P.4, L. 116 is really lengthy. Isn't it a

good idea to make this paragraph more concise and move the rest of it to Supplementary Information for the flow of the to-be-paper?

End of Comments

Masato YOSHIYA

Reviewer #3 (Remarks to the Author):

The authors report on a volume contraction on the formation of hydrides in Au, which is unusual as hydride formation is normally associated with a volume expansion. The authors use computer modeling to interpret the experimental results. Clearly, if the observations are right, they would be of interest to the scientific community.

For this paper to be of interest to a broad range of readers, it will be important for the authors to compare their findings with those of others. For example, there is a significant body of work suggesting that a high concentration of vacancies accompanies the introduction of hydrogen into a metal.

The location of hydrogen in fcc metals is generally expressed as being in either the octahedral or tetrahedral position. A brief literature search shows that a number of computational studies have been performed on the location of hydrogen in the fcc lattice of Au. It is shown that in Au, the tetrahedral site is favored over the octahedral site. Furthermore, if vacancies are introduced into the Au lattice, up to 4 hydrogen atoms can be accommodated around the vacancy. There is still a shift of the center of the hydrogen atom cluster towards the tetrahedral site. The authors are encouraged to review this literature and put their calculations into context.

The authors state that they have introduced some % of hydrogen into the lattice. For metal systems it is common to express the hydrogen concentration in terms of atomic parts per million. Based on the amount of hydrogen that is implanted, what is the estimated size of the hydrides or number of hydrogen atoms in each hydride? There are simple methods to measure the amount of hydrogen that is introduced into a sample, and from the desorption data it is possible to determine the sites at which hydrogen is trapped. This would be different for hydrogen trapped at vacancies compared to in a hydride. The authors are encouraged to measure the H concentration they have implanted into the sample.

In the reporting of the ion implantation, the authors should provide information on the ion energy, ion fluence and ion dose. Given they are implanting hydrogen, the implantation should be described as ion implantation rather than heavy-ion implantation.

What are the SEM images presented in Supplementary Figure 1 supposed to show? Would the authors classify them as being unchanged by the ion implantation? What does TRIM analysis yield for the implantation profile? What is the level of strain that is introduced by the ion implantation?

Page 1. Line 10. The authors should review the work of Zeides, Buckley and Birnbaum on hydrogen in Al. Al exhibits no volume expansion on the addition of hydrogen because it is energetically more favorable for hydrogen to enter the lattice with a vacancy. This study involved a combination of SANS, SAXS, SEM and TEM characterization. The level of hydrogen introduced into the Al was significantly greater than the solubility limit.

Page 1. Line 27. The authors suggest that the Pd-Au hydride is a solid gold hydride, is this correct? Should it not be a Pd-hydride? What is the impact of alloying Pd with Au on the lattice parameter? This system shows a lattice expansion and not a contraction, which is inconsistent with the results presented by the authors. Why the difference if it is a Au hydride?

Page 2 Line 60 This is supposed to be a current density not a current so the units need to be properly expressed. How does the current density used in this work compare to that used in other studies?

Page 3. line 79. What is the meaning of the phrase "... temperature as the metals."

Page 3 line 98 CCG/PBE should be defined

Page 5. line 139. Should "brand" be branches?

REPLY TO REVIEWERS' COMMENTS

Dear reviewers,

Please accept our sincere thanks for your detailed and critical comments, suggestions as well as your great patience of reading through the manuscript. Below are our point-to-point responses provided with your original comments.

Reviewer #1:

The revised manuscript has addressed all the points I had previously raised and thus I suggest the editor to accept the revised manuscript for publication in Nature Communications.

Before formal acceptance, I recommend the authors to reconsider following points.

Thank you very much for the acceptance, your critical comments in the previous round are very valuable for us to improve the quality of the manuscript.

1. "relative effect" vs "relativistic effects". As far as I understand, the latter is "more" appropriate, although the former is not incorrect. The former expression appear in P. 1, L. 32, P. 6, L. 199. Doing a google search using the double-quoted keywords with "site:nature.com" shows the latter is more related to what the authors try to mean in this manuscript.

Thank you for the suggestion, we have changed "relativity effect" to "relativistic effect" through the manuscript (P1, L33 bottom; P7, L214).

2. Readers will find "This result explains why the bulk AuH is not stable under ambient conditions but only at high pressure" at P. 6, L. 200, be an overstatement. This result does not explain all the reason for the stabilization at increased pressure. This is a trivial issue, but I am afraid this overstatement make readers wonder if other solid statements are the same.

We agree and have removed it (P7, L215): the sentence "... these values are still very large. This result explains why the bulk AuH is not stable under ambient conditions but only at high pressure. Applying 100 GPa would..." is changed to "...these values are still very large. Applying 100 GPa would..."

3. Two sequential paragraphs starting at P. 6, L. 197 may confuse many readers except for the ones who themselves frequently do ab initio calculations. The explanations are given based on the model (supercell) size, while most of the readers would understand from the opposite viewpoint: concentration of H. Thus, to avoid interupping the flow of the paper for most of the readers, I suggest the authors to reconsider the description of the supercellsize/concentration dependence of the results.

Thank you, we added the following paragraph to P6, L200: "The nominal H-concentration is straightforward from the size of each supercell, i.e., $100/n^3$ (in %) or $10^6/n^3$ (in ppm). Evidently, diluted concentrations require large supercells to model" and changed the manuscript where applicable to follow with your suggestion (marked in blue).

We also added the values of ppm to concentrations where applicable.

Likewise, it would be more general if the author modified the equation for the cohesive energy to,

$$\Delta E_{n,m} = E(\text{Au}_n\text{H}_m) - [E(\text{Au}_n) + mE(\text{H})]$$

Even if the authors chose to stay in the current way of explanation, explicitly specifying $m=1$ will make it easier to understand for the most of readers. Since this appears in the beginning of Discussion and

following portions are interesting, I hope the most readers give up reading the to-be-paper at this point. Although this is a trivial issue of expression, but it does affect the number of citation after publication.

We put the equation as suggested. This is a great point, thank you again.

4. [Major] To add one sentence each.

In order for me to have objective perspective about the statements in this manuscript, I spent a week after reading the replies to comments before reading the revised manuscript. When I read through the replies, I was immediately convinced. After a week, I could not be fully convinced when I read through the manuscript and needed back to the replies.

Thus, I suggest the authors to add one sentence for each point I raised in the previous round in the beginning or wherever to convince the reviewers that the authors have thoroughly considered the differences between hydride molecules and solid gold, the limitation and sufficiency of Raman spectral analysis in this specific case, clear distinction between static lattice expansion due to static Au-H bonds and dynamic ones due to its extremely light mass of H, and so on. And/or put a stronger pointer to Supplementary Information.

We added the following sentences to the manuscript:

P2, L52: "Therefore, a question arises whether this picture also holds for the bulk gold hydrides Au_nH_m at diluted hydrogen concentrations ($n \gg m$), or there is another kind of dynamic hydride bonding due extreme light mass and high mobility of hydrogen."

P4, L132: "Different bonding configurations are consequences of different local symmetries of hydrogen and naturally, resonances originating from different sets of Au-H bonds contribute differently to phonons."

P6, L178: "It is clear from the analysis given that the hydride formation Au_nH needs be considered in a dynamic manner so that the hydrogen relocation is a primary cause of phonon anharmonicity inducing lattice contraction."

We hope that within the context of the manuscript these additions will make the manuscript clearer and easier to read.

5. For better readability

Paragraph newly added in the revised manuscript starting at P.4, L. 116 is really lengthy. Isn't it a good idea to make this paragraph more concise and move the rest of it to Supplementary Information for the flow of the to-be-paper?

We have half-cut the text, rewrote the removed paragraphs and placed them in the Supplementary Materials file.

End of Comments

Reviewer #3:

The authors report on a volume contraction on the formation of hydrides in Au, which is unusual as hydride formation is normally associated with a volume expansion. The authors use computer modeling to interpret the experimental results. Clearly, if the observations are right, they would be of interest to the

scientific community.

For this paper to be of interest to a broad range of readers, it will be important for the authors to compare their findings with those of others. For example, there is a significant body of work suggesting that a high concentration of vacancies accompanies the introduction of hydrogen into a metal.

Thank you for the suggestion, we appreciate very much this comment. Indeed, there are many studies, both theoretical and experimental ones, regarding the interaction of hydrogen in fcc lattice of metals. However, the ones that include gold are really very few, and we have included them all in the manuscript. It is also not necessary to add citations to all that concerned with hydrogen and metallic structures, so we limit selectively to those containing gold, fcc, and DFT. It is worthwhile noting that, many DFT studies are for surfaces, not for bulk lattices.

The following new references are added (and numbered consecutively as they appear in the manuscript) (P14, L464):

34. Rodbell, K. P., & Ficalora, P. J. *The role of hydrogen in altering the electrical properties of gold, titanium, and tungsten films*. J. Appl. Phys. 65(8), 3107–3117 (1989). <https://doi.org/10.1063/1.342707>
35. Wert C.A. (1978). *Trapping of hydrogen in metals*. In: Alefeld G., Völkl J. (eds) *Hydrogen in Metals II*. Topics in Applied Physics, vol. 29. Springer, Berlin, Heidelberg. https://doi.org/10.1007/3-540-08883-0_24
36. Maeland, A. J. (1968). *A neutron-diffraction study of the α phase in the palladium–gold–hydrogen and palladium–gold–deuterium systems*. Can. J. Phys. 46(2), 121–124 (1968). <https://doi.org/10.1139/p68-017>
37. Ferrin, P., Kandoi, S., Nilekar, A. U., & Mavrikakis, M. *Hydrogen adsorption, absorption and diffusion on and in transition metal surfaces: A DFT study*. Surface Science 606(7-8) 679–689 (2012). <https://doi.org/10.1016/j.susc.2011.12.017>
38. Semidey-Flecha, L., Ling, C., & Sholl, D. S. *Detailed first-principles models of hydrogen permeation through PdCu-based ternary alloys*. J. Membr. Sci. 362(1-2) 384–392 (2010). <https://doi.org/10.1016/j.memsci.2010.06.063>
39. Kristinsdóttir, L., & Skúlason, E. *A systematic DFT study of hydrogen diffusion on transition metal surfaces*. Surf. Sci. 606(17-18) 1400–1404 (2012). <https://doi.org/10.1016/j.susc.2012.04.028>
40. Gómez, E. del V., Amaya-Roncancio, S., Avalle, L. B., Linares, D. H., & Gimenez, M. C. *DFT study of adsorption and diffusion of atomic hydrogen on metal surfaces*. Appl. Surf. Sci. 420, 1–8 (2017). <https://doi.org/10.1016/j.apsusc.2017.05.032>
41. Liu, D., Gao, Z. Y., Wang, X. C., Zeng, J., & Li, Y. M. *DFT study of hydrogen production from formic acid decomposition on Pd-Au alloy nanoclusters*. Appl. Surf. Sci. 426, 194–205 (2017). <https://doi.org/10.1016/j.apsusc.2017.07.165>
42. Züttel, A., Wenger, P., Sudan, P., Mauron, P., Orimo, Shin-ichi. *Hydrogen density in nanostructured carbon, metals and complex materials*. Mater. Sci. & Eng. B 108, 9–18 (2004). <https://doi.org/10.1016/j.mseb.2003.10.087>
43. Manchester, F. D., San-Martin, A., & Pitre, J. M. *The H-Pd (hydrogen-palladium) System*. J. Phase Equilib. 15(1) 62–83 (1994). <https://doi.org/10.1007/bf02667685>

44. Iyer, R. N., & Pickering, H. W. *Mechanism and Kinetics of Electrochemical Hydrogen Entry and Degradation of Metallic Systems*. *Ann. Rev. Mater. Sci.* 20(1), 299–338 (1990).
<https://doi.org/10.1146/annurev.ms.20.080190.001503>
45. Birnbaum, H. K., Buckley, C., Zeides, F., Sirois, E., Rozenak, P., Spooner, S., & Lin, J. S. *Hydrogen in aluminum*. *J. Alloy. Compd.* 253-254, 260–264 (1997).
[https://doi.org/10.1016/s0925-8388\(96\)02968-4](https://doi.org/10.1016/s0925-8388(96)02968-4)
46. Buckley, C., & Birnbaum, H. *Characterization of the charging techniques used to introduce hydrogen in aluminum*. *J. Alloy. Compd.* 330-332, 649–653 (2002).
[https://doi.org/10.1016/s0925-8388\(01\)01496-7](https://doi.org/10.1016/s0925-8388(01)01496-7)
47. Fukai, Y., & Ōkuma, N. *Formation of Superabundant Vacancies in Pd Hydride under High Hydrogen Pressures*. *Phys. Rev. Lett.* 73(12), 1640–1643 (1994).
<https://doi.org/10.1103/physrevlett.73.1640>
48. Ziegler, J. F., Biersack, J. P., Littmark, U. (1985). *The Stopping and Range of Ions in Matter*. New York. Pergamon Press. ISBN 978-0-08-021607-2. See also <http://www.srim.org>

The following text has also added to the Introduction section to discuss the newly added literatures (P2, L56 and P3, L68):

"There are many studies regarding solute hydrogen in *fcc* metals, e.g. in Pd, Cu, Al, but those on hydrogen in gold or in alloyed gold are very limited [34, 35, 36, 37, 38, 39, 40, 41]. It is known that there are α and β phase of hydrogen in metals. The first relates to the exothermal solid solution of atomic hydrogen at low concentration and the second corresponds to the formation hydrides at high hydrogen content [42]. The word hydride here refers to the molecular forms of metal hydrides within the metal lattices. The α phase can also exist in liquid metals under modest hydrogen pressure, e.g. in Pd around 1% [43].

To our concern, there is a report on the lowering of resistivity by H₂ adsorption on Au thin film of 0.5 μm [39]. The authors tried to explain this effect by reduction of lattice strain induced by H₂ chemisorption. However, the impact of lattice strain itself on resistivity is not clear. Furthermore, hydrogen occurs in metals in atomic form, and the main process that happens at the surface of metals is the dissociation of H₂ into atomic hydrogen [44]. So the decrease of resistivity by hydrogen insertion needs an explanation on a more sound basis.

Another important observation is the absence of lattice expansion in H-charged Al reported in Refs.[45, 46]. This effect is assigned to the existence of large amount of vacancies generated during hydrogen charging process. Using the method and data provided in these articles and in Ref.[47] we can estimate the concentrations of vacancies in our samples of order 5–27% (for samples < 0.11% doped). These values are compatible with that of Pd-H (18%) but are sufficiently large in comparison with 0.9% (8 vacancies per ion) estimated by TRIM program [48] for a 100 keV proton beam at 1 μm Au target."

The location of hydrogen in *fcc* metals is generally expressed as being in either the octahedral or tetrahedral position. A brief literature search shows that a number of computational studies have been performed on the location of hydrogen in the *fcc* lattice of Au. It is shown that in Au, the tetrahedral site is favored over the octahedral site. Furthermore, if vacancies are introduced into the Au lattice, up to 4 hydrogen atoms can be accommodated around the vacancy. There is still a shift of the center of the hydrogen atom cluster towards the tetrahedral site. The authors are encouraged to review this literature and put their calculations into context.

Thank you, we have reviewed the literature and added citation and discussion. In particular, the following text was added (P7, L203).

"The available low level theory studies on hydrogen interaction with metallic surfaces usually consider hydrogen in tetrahedral ($\bar{4}3m$) or octahedral ($m\bar{3}m$) site or both [34, 35, 36, 37, 38, 51, 52]. These studies implement either the slab models (i.e. the periodic lattices of limited size with vacuum layers inserted) or the clusters of less than 100 atoms. It is clear that these models cannot apply to the hydrogen trapped deeply inside the large lattice of thousand surrounding atoms, that is, with the cases of diluted concentrations where interactions between hydrogen atoms are negligible and formation of hydride precipitates is not possible. The energies reported by these studies depend heavily on the models involved. The largest binding energy for hydrogen in gold is 2.27 eV [34]."

The authors state that they have introduced some % of hydrogen into the lattice... The authors are encouraged to measure the H concentration they have implanted into the sample.

The 5SDH-2 Pelletron accelerator monitors the ion dose directly via the dose controller (current / charge integrator) at the sample surface. The typical current integration accuracy for a 20nA proton (^1H) beam is $\pm 0.02\%$. The beam current can be set to as low as 1 nA, and with this accuracy it provides dose (ion density) from 1.000×10^{10} to 9.999×10^{18} ions / cm^2 . The charge integration is set (by default) to 0.1 nC / pulse range from 1 to 1000 Hz. Therefore, with knowledge of implantation area and depth (estimated by TRIM program), the H-concentration can be determined with high accuracy. However, we have also measured H-concentration by Gas Chromatography (Aligent) and provide these values together with the ones obtained from charge integration. The method is similar to the one described in [J. Chromatogr. A 1057(1-2), 219-223 (2004)].

These H-concentrations are given in the following file:

"Supplementary_ion_concentration.xls"

For metal systems it is common to express the hydrogen concentration of terms of atomic parts per million.

We have added the values of concentration in appm to the manuscript beside micro-Coulomb which is native to our ion implantation.

Based on the amount of hydrogen that is implanted, what is the estimated size of the hydrides or number of hydrogen atoms in each hydride?

When optimized in various possible symmetries (Table 1s, Supplementary Materials file), the optimized geometries show 4 preferred bonding configurations: $\text{Au}_1\text{-H}$, $\text{Au}_2\text{-H}$, $\text{Au}_3\text{-H}$, and $\text{Au}_4\text{-H}$. We mention this in the manuscript (P7, L226). However, these configurations show only local symmetries of hydrogen atom in the lattice and a set of closest Au-H contacts that contribute mainly to phonons (and some also appear in the Raman spectra of samples). It is clear from our conclusion that the hydride formation in Au:H at dilute concentrations needs be classified as $\text{Au}_n\text{-H}$, where H is bonding to Au_n 6s cloud in a dynamic manner so that its dislocation is a primary cause of phonon anharmonicity inducing lattice contraction. The size of the hydrides in this case should be regarded as the size of the corresponding supercell.

We added the following text directly in the Introduction section to mention our notion of dynamic hydride bonding in metallic gold clearer:

"Therefore, a question arises whether this picture also holds for the bulk gold hydrides Au_nH_m at diluted hydrogen concentrations ($n \gg m$), or there is another kind of dynamic hydride bonding in the lattice, due to extreme light mass and high mobility of hydrogen."

There are simple methods to measure the amount of hydrogen that is introduced into a sample, and from the desorption data it is possible to determine the sites at which hydrogen is trapped. This would be different for hydrogen trapped at vacancies compared to in a hydride.

Energetically hydrogen prefers the tetrahedral position $\bar{4}3m$. This is known from the past and is confirmed in many studies. We show here that the tetrahedral position is not lowest in energy and for hydrogen to be trapped in hydride bonding with Au_n 6s cloud it needs to occupy the P1 positions with small number of close contacts, preferably 2. The occurrences of hydrogen in these positions are shown in the presented Raman spectra and attribute to the binding energy larger than 3.5 eV. Of course, it is interesting to measure the binding energy by mean of other technique, i.e. by desorption or the like. Raman spectroscopy is itself one of possible choices, although it shows energy indirectly, its sensitivity is recognizably high (~ 0.1 meV, or 1 cm^{-1}).

In the reporting of the ion implantation, the authors should provide information on the ion energy, ion fluence and ion dose. Given they are implanting hydrogen, the implantation should be described as ion implantation rather than heavy-ion implantation.

We have updated the Supplementary Materials file for details concerning with ion beam, such as energy, doses and other settings. We have also changed "heavy-ion implantation" to "ion implantation" throughout the manuscript (P1, L7; P3, L78). Thank you very much for the comments.

What are the SEM images presented in Supplementary Figure 1 supposed to show? Would the authors classify them as being unchanged by the ion implantation? What does TRIM analysis yield for the implantation profile? What is the level of strain that is introduced by the ion implantation?

The aim of SEM image is to show the visual difference (darker surface) of implanted sample. After implantation the implanted area can easily be recognized by the change of its color in contrast to the surrounding areas.

We added comment to SEM images (in Supplementary Materials file).

The TRIM program shows that at 100 keV the proton beam cannot penetrate through a 700 nm thick Au film with 99.03% energy loss for ionization, 0.07 for recoils ionization; 0.25% for phonons; 1.65% backscattered, 8 vacancies per ion, 0.03% transmitted; ion stop range of 3928 nm; std. dev. 1162 nm.

We have updated the Supplementary Materials file accordingly.

Page 1. Line 10. The authors should review the work of Zeides, Buckley and Birnbaum on hydrogen in Al. Al exhibits no volume expansion on the addition of hydrogen because it is energetically more favorable for hydrogen to enter the lattice with a vacancy. This study involved a combination of SANS, SAXS, SEM and TEM characterization. The level of hydrogen introduced into the Al was significantly greater than the solubility limit.

Thank you. We have included the citation to the works of Zeides, Buckley and Birnbaum in the list of references.

As mentioned above, the following texts are added to discuss the works:

"Another important observation is the absence of lattice expansion in H-charged Al reported in Refs.[45, 46]. This effect is assigned the existence of large amount of vacancies generated during hydrogen charging process. Using the method and data provided in these articles and in Ref.[47] we can estimate the concentrations of vacancies in our samples of order 5÷27% (for samples < 0.1% doped). These values are compatible with that of Pd-H (18%) but are sufficiently large in comparison with 0.9% (8 vacancies per ion) estimated by TRIM program [48] for a 100 keV proton beam at 1µm Au target."

The progressive generation of vacancies may cause the lattice contraction indeed. But this effect should be linear, that is, the contraction should increase with the number of vacancies and grows with hydrogen content. This means we should observe the lattice contraction at high H doses, but no such observation has been reported until now. In contrast, we show here that the lattice contracts only at low hydrogen concentration while it expands at high concentration, and the metallic form of hydride bond Au_n-H occurs only in diluted region.

Page 1. Line 27. The authors suggest that the Pd-Au hydride is a solid gold hydride, is this correct? Should it not be a Pd-hydride? What is the impact of alloying Pd with Au on the lattice parameter? This system shows a lattice expansion and not a contraction, which is inconsistent with the results presented by the authors. Why the difference if it is a Au hydride?

We are sorry, we do not claim they are the gold hydrides, at most they are the solid hydrides containing gold.

P1, L26: the sentence may be mis-leading, so we have modified it to read now:

"These Pd-Au alloys are the only forms of solid hydrides containing gold that are available today at ambient pressure. The known solid gold hydrides [21] are unstable and could be prepared only at high temperature and pressure."

These Pd-alloys are heavily doped with hydrogen in contrast to our samples which contain only small amount of hydrogen (<0.11%). From the available literatures [6, 7, 8, 9, 10] we do not know about the role of hydrogen in these compounds (position, lattice, binding, symmetry, clustering, configuration, etc...). But we explain in our manuscript why the host metallic lattice should contract at low doses and expand at higher ones. The Pd hydrides may express the same.

Page 2 Line 60 This is supposed to be a current density not a current so the units need to be properly expressed. How does the current density used in this work compare to that used in other studies?

Thank you, we changed the unit to "nanoampere per cm²" (P3, L85). For comparison, the beam current density used in Ref. 6 (Pd-alloys) is 4 µA/cm². The beam in our case is maximal 20 nA/cm².

Page 3. line 79. What is the meaning of the phrase "... temperature as the metals."

We mean the samples are not metallic at room temperature, so we have rewritten the sentence to make it clear. The whole sentence is now:

"Indeed, for the heavily doped samples previously reported [22, 23], a drop in the conductivity due to implantation was observed, and the samples were not metallic at room temperature."

Page 3 line 98 CCG/PBE should be defined

We are sorry for this mistake, it should be GGA/PBE and means the Perdew-Burke- Ernzerhof Generalized Gradient Approximation (Phys. Rev. Lett. 77, 3865 (1996)). Since this functional is very common today we only added citation to it in the Supplementary Materials file.

Page 5. line 139. Should "brand" be branches?

Thank you very much for this, it is a mistake and we have corrected it (P7, L171, L174).

We would like to express our deep gratitude to both reviewers for their critical, valuable comments and suggestions, which helped us a lot to improve the quality of this manuscript. We also send our warm thanks to the editors for handling of the manuscript and timely communication and assistance.

Sincerely yours

REVIEWERS' COMMENTS

Reviewer #3 (Remarks to the Author):

The authors have appropriately addressed the previous comments for the most part.

The authors may wish to review again the papers by Buckley et al. on hydrogen in Al. From my review of that work, the vacancies were not derived from the charging condition per se but because it was energetically more favorable for hydrogen to enter the bulk material in association with a vacancy. This is an important point as some hydrogen charging mechanisms, notably electrochemical charging at high current densities, do introduce damage to the surface region of the metal. Consequently, materials charged using this method have a high hydrogen concentration trapped in the damaged surface. This is not how I interpret the work of Buckley and colleagues. They also used several different charging methods to eliminate the possibility of the vacancies being the result of the charging condition.

REPLY TO REVIEWERS' COMMENTS

Reviewer #3 (Remarks to the Author):

The authors have appropriately addressed the previous comments for the most part.

The authors may wish to review again the papers by Buckley et al. on hydrogen in Al. From my review of that work, the vacancies were not derived from the charging condition per se but because it was energetically more favorable for hydrogen to enter the bulk material in association with a vacancy. This is an important point as some hydrogen charging mechanisms, notably electrochemical charging at high current densities, do introduce damage to the surface region of the metal. Consequently, materials charged using this method have a high hydrogen concentration trapped in the damaged surface. This is not how I interpret the work of Buckley and colleagues. They also used several different charging methods to eliminate the possibility of the vacancies being the result of the charging condition.

Dear reviewer,

Thank you very much again for discussions and suggestions. We fully agree with you that it is energetically more favorable for hydrogen to enter the vacancies. But vacancies are of different nature, and they differ not only in energy but also in symmetry. Some vacancies, when occupied by hydrogen, may cause host atoms to dislocate and reduce lattice symmetry accordingly.

For the case of hydrogenated aluminum, the Al electronegativity is only 1.61 compared to 2.2 of hydrogen, so a hydride bond draws electrons from the conduction band but the outflow would be small. The computation shows that only ~ 0.1 e would be added to hydrogen. Therefore, even if hydrogen enters the most favorable vacancies in aluminium, the weak hydride bond means a weak effect on the lattice. The weak bond also explains why hydrogen occurs favorably at the surface, and when it enters the bulk under external stimuli, damages often occur. These damages can cause the lattice changes of such large magnitude that compensate for and suppress the small expansion caused by hydrogen insertion in regular lattice spaces. However this is only a general remark, we think that each case is different and a through investigation with accurate data and measurements would be needed.

Thank you again, we read your comment carefully and discuss between us, and hope that we have understood your suggestion correctly.

Sincerely yours